# AI-Enabled Crop Management Framework for Pest Detection Using Visual Sensor Data

**DOI:** 10.3390/plants13050653

**Published:** 2024-02-27

**Authors:** Asma Khan, Sharaf J. Malebary, L. Minh Dang, Faisal Binzagr, Hyoung-Kyu Song, Hyeonjoon Moon

**Affiliations:** 1Department of Computer Science and Engineering, Sejong University, Seoul 05006, Republic of Korea; asmakhan28@sju.ac.kr; 2Department of Information Technology, Faculty of Computing and Information Technology, King Abdulaziz University, P.O. Box 344, Rabigh 21911, Saudi Arabia; smalebary@kau.edu.sa; 3Department of Information and Communication Engineering and Convergence Engineering for Intelligent Drone, Sejong University, Seoul 05006, Republic of Korea; minhdl@sejong.ac.kr (L.M.D.); songhk@sejong.ac.kr (H.-K.S.); 4Department of Computer Science, Faculty of Computing and Information Technology, King Abdulaziz University, P.O. Box 344, Rabigh 21911, Saudi Arabia; fbinzagr@kau.edu.sa

**Keywords:** convolution neural network, deep learning, sustainable agriculture, UAV technology, computer vision, monitoring system

## Abstract

Our research focuses on addressing the challenge of crop diseases and pest infestations in agriculture by utilizing UAV technology for improved crop monitoring through unmanned aerial vehicles (UAVs) and enhancing the detection and classification of agricultural pests. Traditional approaches often require arduous manual feature extraction or computationally demanding deep learning (DL) techniques. To address this, we introduce an optimized model tailored specifically for UAV-based applications. Our alterations to the YOLOv5s model, which include advanced attention modules, expanded cross-stage partial network (CSP) modules, and refined multiscale feature extraction mechanisms, enable precise pest detection and classification. Inspired by the efficiency and versatility of UAVs, our study strives to revolutionize pest management in sustainable agriculture while also detecting and preventing crop diseases. We conducted rigorous testing on a medium-scale dataset, identifying five agricultural pests, namely ants, grasshoppers, palm weevils, shield bugs, and wasps. Our comprehensive experimental analysis showcases superior performance compared to various YOLOv5 model versions. The proposed model obtained higher performance, with an average precision of 96.0%, an average recall of 93.0%, and a mean average precision (mAP) of 95.0%. Furthermore, the inherent capabilities of UAVs, combined with the YOLOv5s model tested here, could offer a reliable solution for real-time pest detection, demonstrating significant potential to optimize and improve agricultural production within a drone-centric ecosystem.

## 1. Introduction

The agricultural sector is vital to economic enhancement, and it is essential to identify the pests that harm it. Pest detection, a persistent challenge, leads to a substantial annual loss of 20% in global crop yields [1,2]. Timely detection of plant diseases and pests is critical for efficient agricultural output, as shown in Figure 1. Embracing UAV technology becomes important for a more efficient and technologically driven approach to safeguard crop yields, including enhanced crop monitoring using UAVs [1,2]. This process significantly impacts grain yield, agricultural progress, and farmers’ income [3]. Additionally, high-resolution data for yield prediction further enhances the precision of our approach. To tackle the aforementioned challenge, an artificial intelligence (AI)-driven model stands out as the optimal choice, playing a pivotal role in advancing modern agricultural research. Leveraging the inherent capabilities of this cutting-edge technology, pest detection, and classification are seamlessly executed with exceptional efficiency, facilitating prompt intervention in agricultural production. The effectiveness of this approach not only hints at a reduction in losses but also promises a significant boost in agricultural production, especially when integrating UAV technology for heightened precision and real-time monitoring [4]. Detecting and preventing crop diseases are crucial aspects of this innovative approach, contributing to overall agricultural resilience and productivity. To this end, researchers have explored both ML- and DL-based modelsfor wheat diseases and pest recognition [5,6].

Plant analysis laboratories relying on morphological features for pest identification are limited because taxonomy specialists need to perform accurate classifications [7]. However, these approaches for pest identification have certain limitations, including the fact that the accurate classification of pests requires experts in the field of taxonomy, possible human error, and difficulties in identifying pests quickly.

Several methods have been proposed for automatic pest detection using traditional ML [8]; for example, Faith et al. [9] used manual feature extraction and relative filters to identify different pest species using K-means clustering algorithms. However, this approach is very time-consuming, particularly for large datasets. Next, Rumpf et al. [10] presented a method based on support vector machines and spectral vegetation-based detection of sugar beet disease. Pests can be detected using these methods; however, they have several limitations, including inefficiency when manual feature extraction is required, making them time-consuming, tedious, error-prone, and dependent on computer experts. Later, owing to the inadequate feature extraction process, researchers diverted to DL, characterized by multilayer neural networks that enable automated end-to-end feature extraction has emerged as an alternative solution. This shift toward DL improves recognition efficiency while minimizing the lengthy manual feature extraction process [11]. 

A DL-based technique for pest and disease detection in tomato leaves was developed by Shijie et al. [12] with an average accuracy of 89%. However, this method is limited to identifying pests with a simple background and is difficult to implement in real-time applications. Generative adversarial network augmentation was utilized by Gandi et al. [13] to create an efficient DL model for categorizing plant diseases. Leonardo et al. [14] discussed the economic importance of fruit flies in Brazil and the challenges associated with identifying these pests. Later, they applied DL and transfer learning techniques that achieved an impressive accuracy of 95.68%. Transfer learning was also used by Dawei et al. [15] to detect ten pest species with 93.84% accuracy. In contrast, DL pest classification models performed well and boosted the existing discriminative score. However, challenges such as deplorability over resource-constrained devices, robustness concerns, lower accuracy, and high equipment costs hinder the integration of existing DL approaches into real-world scenarios. Considering these challenges, this study presents a novel approach to efficient pest detection using a modified YOLOv5 model. Our model performs better than object recognition models due to its fast inference speed, high mAP, robust adaptation, and higher accuracy. Moreover, the integration of high-resolution data for yield prediction enhances the overall reliability of our model.

In summary, this study represents a significant step towards efficient pest detection and has profound implications for sustainable agriculture. The proposed model has the potential to revolutionize pest management by maximizing agricultural yields while mitigating losses. We aim to pave the way for a more resilient and productive agricultural future by combining cutting-edge methods and custom enhancements. The major contributions of the proposed model are summarized below.

We present an advanced system that uses UAVs to identify pests in real time. This groundbreaking method surpasses previous approaches with enhanced accuracy and a notable reduction in false alarms. By incorporating UAV technology, we have achieved a significant improvement in pest detection, highlighting the effectiveness of merging UAVs with this innovative solution.We refined the internal architecture of YOLOv5s by replacing smaller kernels in SSP (Neck) with larger ones and introducing a Stem module into the backbone. This strategic modification enhances the model’s capability to efficiently identify pests of varying sizes in images, reducing time complexity. Through extensive experimentation and comparison with nine object-detection models using a pest detection dataset, our model demonstrated empirical effectiveness and outperformed existing methods. A qualitative assessment further solidified the superior performance of our UAV-assisted pest detection technology.

The rest of the paper is structured as follows: Section 2 provides a comprehensive summary of recent research relevant to the topic of a modified YOLOv5s architecture for pest and insect detection, and Section 3 outlines the data collection and methodology employed for the system. The experimental results are presented and analyzed in Section 4 and Section 5 present the conclusions drawn from this research.

## 2. Literature Review

To cope with the aforementioned challenges, therefore, several researchers have been working on automatic systems for detecting insects in sustainable agriculture. Cheeti et al. [16] made a significant addition to this field when they used cutting-edge methods to categorize and detect pests in sustainable agriculture, including convolutional neural networks (CNN). Notably, their research involved creating a dataset using data from Internet sources, which yielded promising performance. To detect rice diseases and insects with an accuracy of 90.9%, Mique et al. [17] used CNN and image processing. They also released their suggested approach as an application for mobile devices for public use. However, this method is expensive in terms of computations and needs to be more accurate. A single-shot multi-box detector (SSD) with fine-tuning procedures was developed by Nam et al. [18] to recognize and classify collected insects. Academic research was the focus of a thorough comparative study by Burhan et al. [19]. In agricultural environments, their study concentrated on using four trained deep-learning models to identify diseases and detect pests in rice-growing regions. With an accuracy rating of 86.799%, their model showed good performance. Nonetheless, further improvements are necessary to enhance the model’s performance across evaluation metrics.

Kouba et al. [20] used accelerometer sensors to create a unique dataset as part of a sensor-based approach to agricultural monitoring. Their system is also integrated into a mobile application accessible to the public, allowing for the early detection of red palm weevil (RPW) through movement analysis. However, their system is based on voice and movement analysis, which has a higher false-positive rate. Habib et al.’s [21] model for identifying and categorizing brown- and yellow-rusted illnesses in wheat crops uses classical machine learning. To help coffee growers, Esgario et al. [22] created a mobile app and a CNN model specifically designed to identify biotic stresses in coffee leaves. Svenning et al. [23] introduced a pre-trained CNN model along with fine-tuning techniques to classify carabid beetle species. Of the test images, 51.9% were correctly classified to species level, giving an average classification rate of 74.6% due to their efforts. Nonetheless, the model’s testing phase speed has impeded its feasibility for real-time implementation. The Deep-PestNet model, which has eight convolutional layers and three fully connected layers for effective pest detection, has been introduced by researchers [24]. However, the approaches have several limitations, they used CNN methods in the pest detection domain, which mainly focus on classification. These methods classify the entire image as a single class that does consider the fact that pests or insects typically occupy a small portion of an image. When the object features are not prominently visible, relying solely on the complete image feature, without region proposals, can lead to reduced detection performance. Moreover, the CNN methods come with a high computational complexity, rendering them unsuitable for practical real-world implementation.

Therefore, researchers have used DL-based object-detection models to overcome the limitation of the CNN-based methods. For instance, Li et al. [25] utilized the IPI02 dataset to detect insects in fields using various DCN networks, including Faster-RCNN, Mask-RCNN, and YOLOv5. They obtained promising results and demonstrated that YOLOv5 outperformed Faster-RCNN and Mask-RCNN, which attained 99% accuracy, whereas YOLOv5 gained 97%. Hu et al. [26] used YOLOv5 and near-infrared imaging technologies to classify and detect pests in agricultural landscapes accurately, and they were able to do so with a significant mean average precision (mAP) of 99.7%. Similarly, Chen et al. [27] suggested an AI mobile-based approach utilizing a particular dataset, especially for pest identification in agricultural areas.

To achieve a high identification score, they examined various pre-trained DL models, including YOLOv4, single-shot detectors (SSDs), and faster R-CNNs. Notable achievements include a 100% F1 score for mealybugs, an 89% F1 score for coccidia, and a 97% F1 score. YOLOv4 has also demonstrated remarkable performance in terms of F1 score. Legaspi et al. [28] developed a YOLOv3 model for identifying pests, such as fruit flies and white flies, using hardware alternatives, such as Raspberry Pi, desktop, and online apps accessible to the general population. Their model achieved 83.07% accuracy for pest classification and detection but required additional refinement for more accurate predictions. Furthermore, using a smartphone application, Karar et al. [29] presented a DL method, with the Faster-RCNN architecture outperforming other advanced DL models, and achieved a remarkable 99.0% accuracy for pest detection in agricultural fields. To identify and categorize red palm weevils (RPW), Alsanea et al. [30] developed a region-based CNN that performs best in evaluation matrices when utilizing the RPW dataset. The model’s complexity and speed of inference make it hard to implement in real time. Using a customized dataset, Liu et al. [31] developed a YOLOv3-based DL model for tomato disease and pest identification in realistic agricultural environments. 

In the above-mentioned method, Faster-RCNN has a large number of learning parameters and model size, which restrict the model from real-world implementation. In the context of the YOLO base model, there is a need for further improvement in deep-learning-based object detection to detect small objects in complex backgrounds with higher inference speeds [32]. Furthermore, YOLOv5 models have demonstrated effectiveness in object detection, but their conventional versions may lack optimal efficiency and precision in the context of pest detection, particularly when applied to UAV-based monitoring in agricultural settings. The inherent challenges of agricultural environments, such as diverse crop types, varying terrains, and the need for real-time monitoring, can strain the performance of traditional YOLOv5 models. These models may not be fully adapted to the intricacies of pest identification within the dynamic and variable conditions present in agricultural fields. As a result, there is a potential for reduced accuracy and reliability in pest classification, limiting their practicality for comprehensive and real-time pest management solutions. This underscores the necessity for tailored optimizations, as presented in our study, to enhance the efficacy of YOLOv5 models in addressing the specific demands of UAV-centric pest detection systems in sustainable agriculture.

## 3. The Proposed Methodology

Based on insights from the existing literature, we adapted a modified YOLOv5s model to effectively identify the region of interest in each image and subsequently assign appropriate class labels to them. The entire steps followed by dataflow of the proposed model is given in Figure 2, while technical detail is provided in the following subsections.

### 3.1. The Proposed Pest Detection Model

Object-detection models play a crucial role in identifying and categorizing specific regions in an image. These models are widely used in various areas of computer vision, owing to their performance and efficiency. However, selecting object recognition models for specific areas can be challenging when determining the exact location of objects and assigning the correct label while saving computational resources. Several studies have been conducted to construct object identification models to modify the YOLO-based model, which has shown remarkable progress. The original YOLO approach, initiated by Redmon et al. [33], addresses object detection as a regression challenge rather than classification. This approach detects target objects using a single neural network, resulting in impressive performance and real-time inference capabilities. In addition, YOLO-based object recognition models show exceptional generalizability, as they quickly adapt to recognize a wide range of objects through training. Enhanced crop monitoring using UAVs, high-resolution data for yield prediction, and detecting and preventing crop diseases are integral components that enhance the efficacy of our proposed approach. The YOLO architecture has recently undergone several improvements to boost its efficacy and efficiency for insect identification. From 2016 to 2018, Redmon and Farhadi [34] proposed the initial three versions of the YOLO-based object-detection models, which attracted several researchers due to its fast inference speed and model accuracy YOLOv4, showing an impressive average accuracy of 43.5% on the MS-COCO dataset while maintaining high speed, was introduced in 2020 by Bochkovskiy et al. [35]. Additionally, YOLOv5 was introduced by Feng et al. [36], achieving a remarkable performance and speed turning point. 

There are five variations of YOLOv5, with diverse feature map depths and breadths: YOLOv5x, YOLOv5l, YOLOv5m, YOLOv5s, and YOLOv5n. A thorough evaluation of the MS-COCO dataset underscored the exceptional results of these models. YOLOv5n excelled in fast inference, whereas YOLOv5x exhibited the highest object-detection accuracy. These models have common structural elements, including input, backbone, neck, and prediction components [37]. Compared to previous YOLO iterations, YOLOv5 provides higher recognition accuracy, lower computational complexity, and a smaller model footprint, making it optimal for resource-constrained devices. This study leveraged YOLOv5’s improved recognition capabilities and real-time inference to refine its inherent architecture and to enable efficient and reliable insect recognition.

### 3.2. Network Architecture

This study used the YOLOv5s object identification system as our primary model because of its quick inference and outstanding performance. The goal of this study was to optimize this system for pest or insect detection. To this end, we made several modifications to the YOLOv5s model to identify small and large insects in the image effectively. The input, backbone, neck, and head modules comprise the modified model’s four essential components. The three sub-modules comprise the input module: imagine scaling, adaptive anchor-box computation, and pre-processing of the mosaic data. The pre-processing phase of the mosaic data includes three techniques: random scaling, cropping, and order, which introduce variability in the positioning of image segments that improve the network’s ability to detect smaller objects, such as insects. The various modules architecture used in the proposed model are given in Figure 3. 

To improve the recognition accuracy for small objects and expand the range of available data, we randomly select the fusion point for merging images. During model training, YOLOv5s dynamically generates multiple prediction boxes based on the initial anchor box. Non-maximum suppression (NMS) was also used to determine which prediction box resembled the original box the most. We continually resized the adaptively zoomed picture before entering it into the network for identification, removing any potential inconsistencies resulting from different image sizes and ensuring compatibility with the feature tensor and the fully connected layer. The YOLOv5 backbone architecture uses a CSPNet backbone based on the Dense Net architecture and easily integrates the focus module. The focus module is a critical component that performs down-sampling by decomposing the input image, which is initially 640 × 640 × 3, and then concatenated to produce a 320 × 320 × 12 feature map. However, our analysis of YOLOv5 models that have already been trained indicates that the focus layer struggles to effectively capture spatial information from tiny objects, which impacts model performance. We suggest adding a Stem unit following the focus module to overcome this restriction, as shown in Figure 2. The Stem module facilitates the creation of more sophisticated feature maps by providing additional down-sampling with a stride of two and an increase in channel dimensions. The Stem module significantly enhances display possibilities while just slightly increasing processing complexity.

In the YOLOv5 model, the CSP module is constructed using a series of 1 × 1 and 3 × 3 convolutional layers. This module divides the initial input into two segments, each passing through its processing path. The first segment is fully processed by a CBS block consisting of convolution, normalization of batch size, and activation functions of SILU, as in the ingenious work of Elfwing et al. [38]. The second segment passes through a convolutional layer, as shown in Figure 1. A CBS module is then added after merging the two partitions. In addition, the spatial pyramid pooling (SPP) block plays an important role in expanding the receptive field and extracting essential features, which are then proficiently passed to the feature aggregation block. This adaptation aimed to improve the ability of the network to identify insects by extracting relevant features from smaller objects, ultimately leading to better insect detection performance. In contrast to using multiple CNN models that require fixed-size input images, the integration of SSP results in the generation of a fixed-size output. This approach also facilitates the acquisition of important features by pooling the different scales of the same module.

In our modified version, the SSP block replaces traditional sizes of the kernel, such as 5 × 5, 9 × 9, and t3 × 13, with 3 × 3, 5 × 5, and 7 × 7, as shown in Figure 2. To further improve accuracy, The input image’s longer edges normalized to 640 pixels, whereas the smaller edge was adjusted appropriately. Furthermore, the shortest edge coincided with the maximum step size of the SPP block. In the absence of P6, the shortest edge had a factor of 32. If P6, however, the shorter edge must be a factor of 64. The neck component plays a critical role in integrating the feature maps generated by various folds in the spine, preparing them for the head segment. According to Hu et al. [39], the neck segment has PAN and FPN structures to improve the network’s feature fusion capabilities. The PAN technology is typically used by YOLOv5, producing three output features: P3, P4, and P5, with dimensions of 80 × 80 × 16, 40 × 40 × 16, and 20 × 20 × 16. But, as you can see in Figure 2**,** we added an extra P6 output block to our customized model, with a 10 × 10 × 16-pixel feature map. The ability of the model to identify both large and small objects in the input image is greatly enhanced by this addition, which was previously employed for face detection by Qi et al. [38]. Lastly, convolutional layers are used in the head component to identify objects by defining bounding rectangles around the regions of significance and then classifying them. Our model incorporated a comprehensive scaling method that uniformly modified the backbone, neck, and head’s resolution, width, and depth. This all-encompassing modification guarantees higher accuracy and efficiency for insect detection.

## 4. Experiments and Results

This section provides a comprehensive overview of the experimental setup, evaluation parameters, dataset selection, model performance, and comparison with state-of-the-art methods.

### 4.1. Experimental Setup

To implement the proposed work, we employed PyTorch with CUDA support to analyze pest and insect detection results. Our hardware setup included an NVIDIA (GeForce RTX 3070 GPU) with 32 GB of RAM. We evaluated our model with various hyperparameter settings, and the optimal performance was obtained with the following configuration: a batch size of 32, the SGD optimizer with a learning rate of 0.001, and training for 200 epochs. 

The proposed model is evaluated using well-known evaluation metrics such as precision, recall, and mAP [40] and considered state-of-the-art evaluation metrics in the target domain.

Precision is an evaluation metric used in ML and DL to assess model performance. It is defined as the ratio of true-positive samples to the sum of true-positive and false-positive samples, as expressed in Equation (1).
(1)$$P=\frac{TP}{TP+FP}$$

Recall is another evaluation metric that focuses solely on positive samples within a given dataset, excluding negative ones, as formulated in Equation (2).
(2)$$R=\frac{TP}{TP+FN}$$

The mAP metric is used to assess object-detection models. It computes a score based on the relationship between the localized boxes and ground-truth bounding boxes. Achieving the highest score is an indicator of an accurate detection approach, as formalized by Equation (3).
(3)$$mAP=\frac{1}{N}\sum_{i-1}^{N}AP_{i}$$

In the equations, TP represents the number of correctly identified positive samples, while FP signifies the number of negative samples that are falsely identified as positive. Similarly, FN corresponds to the number of misclassified positive samples.

### 4.2. Dataset Selection

The data-collection method is essential for the efficacy of model training in artificial intelligence. This information is meticulously organized into five taxonomic groups: Ants, Grasshoppers, Palm Weevils, Shield Bugs, and Wasps. The classes were organized as follows: the ant class contained 392 images, the grasshopper class had 315 images, and the palm weevil, shield bug, and wasp categories had 148, 392, and 318 images, respectively. We used 70% of the dataset for training, 20% for validation, and 10% for testing. Figure 4 demonstrates the total number of instances used to train the proposed model.

The dataset was carefully annotated in accordance with the object recognition model [6] using a publicly available annotation tool from GitHub (accessed on 23 June 2022). The annotation process was written in Python with the cross-platform Qt GUI (graphical user interface) toolkit. To match the criteria of the YOLO-based model, the data collection was annotated in YOLO format, which requires annotated files to be in .txt format. The dataset is divided into three sub-categories training, validation, and testing which contain 70%, 20%, and 10% of the data, respectively. Figure 5 shows samples of the pests from each dataset class. 

### 4.3. The Proposed Model Evaluation

In this subsequent section, we provide a detailed description of the training process for our model. We used an early stopping algorithm to halt the training once the model has reached a certain level of performance and can no longer learn efficiently and effectively. The evaluation of the proposed model includes the examination of various metrics such as loss, recall, precision, and mAP, as depicted in Figure 6.

The loss graph visually represents the model’s capability to identify objects accurately, indicating its proficiency in performing tasks effectively. The object loss function assesses the model’s ability to complete tasks within relevant regions of interest, and an increase in accuracy corresponds to a decrease in the loss function. Accurate and effective object classification relies on reducing associated losses. In Figure 6, Bounding box loss, class loss, and distribution focal loss in the training phase are reached to a minimum of 0.50, 0.02, and 1.0, respectively. Similarly, these metrics in the validation phase reached 1.1, 0.50, and 1.95, respectively. In the context of ML and ML, recall and precision are important metrics for assessing model performance. Elevated precision and recall, as observed in Figure 6, signify enhanced model accuracy. The loss function value consistently decreases during the training process, and the model demonstrates a continuous ability to reduce loss and rapidly improve recall, and precision within a few epochs. At the last epoch, the precision and recall in the training phase reached 0.96 and 0.93 as shown in Figure 6.

Furthermore, the maximum accuracy, recall, and mAP values are achieved at approximately the 120th epoch, highlighting the efficacy of our model. Therefore, the proposed model proves to be a robust and capable solution across various evaluation metrics.

We conducted a performance evaluation of the proposed model using the test set of the dataset, employing a confusion matrix for in-depth analysis, as shown in Figure 7. This confusion matrix encompasses five distinct pest classes and introduces an additional category, “background FN”, to emphasize scenarios where the model failed to identify objects within the image. To present a comprehensive insight into the confusion matrix, we closely analyzed the accuracy in predicting various pest classes, including ants, grasshoppers, palm weevils, shield bugs, and wasps. The accuracy values for these categories were 0.79, 1.00, 0.78, 0.96, and 0.86, respectively. 

### 4.4. Comparative Analysis with State-of-the-Art Models

In this subsequent section, we provide a comprehensive comparison of the proposed model with nine other state-of-the-art models. The proposed model consistently demonstrates superior performance in pest detection, particularly excelling in the identification of ants, grasshoppers, palm weevils, shield bugs, and wasps. The results are summarized in Table 1, indicating that the Faster-RCNN emerges as a powerful contender, achieving promising performance. However, the proposed model surpasses Faster-RCNN with higher average precision, recall, and mAP values of 0.04, 0.03, and 0.3, respectively.

YOLOv3 and YOLOv4 exhibit relatively weaker performance compared to other models. YOLOv5n falls short, with average values of 0.87, 0.88, and 0.89 for precision, recall, and mAP. YOLOv5s achieves 0.91 accuracy, 0.83 recall, and 0.90 mAP, while YOLOv5m outperforms YOLOv5s models, obtaining an average precision value of 0.94, recall of 0.84, and mAP of 0.91. YOLOv5l and YOLOv5x exhibit better performance, as given in Table 1. The EPD model also outperforms other models, but the proposed model surpasses the EPD with higher precision, recall, and mAP values of 0.02, 0.03, and 0.1, respectively. Therefore, Table 1 highlights that our proposed model achieves superior performance across all evaluation metrics.

### 4.5. Splitting Dataset Using 5-Fold Cross Validation

In this strategy, the whole dataset is utilized for training and validation by making five different folds to effectively evaluate the model performance. Figure 8 shows that the proposed model obtained optimal performance for each fold in terms of precision, recall and mAP. The proposed model obtained precision values for each fold: Fold 1: 96.88%, Fold 2: 96.34%, Fold 3: 96.16%, Fold 4: 95.4a 5%, and Fold 5: 95.17%. The recall value of 93.80%, 93.45%, 93.10%, 92.53%, and 92.12% for Fold 1, Fold 2, Fold 3, Fold 4, and Fold 5, respectively. Similarly, the proposed model achieved higher mAP of 95.83%, 95.42%, 95.00%, 94.68%, and 94.07%, respectively. In conclusion, our model obtained an average precision, recall, and mAP of 96.00%, 93.00%, and 95.00%, respectively.

### 4.6. Model Complexity Analysis

Table 2 provides a comprehensive assessment of the proposed model’s feasibility, calculating giga floating point operations per second (GFLOPs), model size, and FPS for all models, which are then compared to the suggested approach. In Table 2, the proposed model is compared with YOLOv5n, YOLOv5s, YOLOv5m, YOLOv5l, and YOLOv5x. In the comparison, the YOLOv5n model exhibited faster inference speed with reduced GFLOPs and model size. However, the YOLOv5n model obtained poor performance as shown in Table 1. In contrast, when compared with YOLOv5s, YOLOv5m, YOLOv5l, and YOLOv5x, the proposed model obtained significantly higher FPS, with 1.50, 12.59, 16.13, and 18.85 times the speed, respectively. This underscores that our model combines high performance with fast inference speed while maintaining a favorable balance of GFLOPs and model size. The results obtained show the efficacy of our model, suggesting that it is a viable solution for real-world application.

### 4.7. Visual Result of the Proposed Model

We performed a visual analysis of the output predictions of the suggested model to evaluate how robust it was. The results of the model for the localization and identification of five different insects or pests are shown in Figure 9. This figure clearly shows how well our model selects regions of interest and correctly labels classes. It performs admirably in both detection and recognition tasks, accurately delineating bounding boxes around objects and precisely labeling their classes. This visual representation highlights the model’s potential for real-time applications. However, it is worth noting that there are some limitations to the proposed approach, as highlighted in Figure 9. While most images are accurately classified, exceptions exist. For instance, the first and last images in the fifth row present recognition challenges. Additionally, instances of misdetection are observed in the second and third images of the second row, as well as in the last image of the third row. These visual results offer valuable insights into the model’s performance, acknowledging its strengths while identifying areas where improvement may be needed.

### 4.8. Discussion

Our study addresses the challenge of crop diseases and pest infestations in agriculture by leveraging UAV technology for enhanced crop monitoring through UAVs and improving the detection and classification of agricultural pests. Conventional methods often involve laborious manual feature extraction or computationally intensive DL techniques. To overcome these limitations, we present an optimized model specifically tailored for UAV-based applications. Our modifications to the proposed YOLOv5s model incorporate advanced attention modules, expanded cross-stage partial network modules, and refined multiscale feature extraction mechanisms, enabling precise pest detection and classification. Inspired by the efficiency and adaptability of UAVs, our research aims to transform pest management into sustainable agriculture while also combating crop diseases.

We conducted extensive experiments on a medium-scale dataset, identifying five agricultural pests: ants, grasshoppers, palm weevils, shield bugs, and wasps. Our thorough experimental analysis demonstrates superior performance compared to various Faster-RCNN [41,42] and YOLO model versions [43,44]. 

Compared with existing methodologies, our model demonstrates competitive performance. For instance, while Faster-PestNet [41] achieves an accuracy of 82.43% on the IP102 dataset. Similarly, Pest-YOLO [43] and PestLite [44] achieve mean average precision scores of 73.4% and 90.7%, respectively. Jiao et al. [45] integrated an anchor-free convolutional neural network (AF-RCNN) with Faster-RCNN for pest detection on the Pest24 dataset, yielding an mAP and mRecall of 56.4% and 85.1%, respectively. Wang et al. [46] employed four detection networks, including YOLOv3, SSD, Faster-RCNN, and Cascade-RCNN, for detecting 24 common pests, with YOLOv3 achieving the highest mAP value of 63.54%. Furthermore, AgriPest-YOLO achieves a mean average precision (mAP) of 71.3% on a multi-pest image dataset by integrating a coordination and local attention mechanism, grouping spatial pyramid pooling fast, and soft non-maximum suppression, facilitating efficient and accurate real-time pest detection from light-trap images [47].

The proposed model achieved higher performance metrics, with an average precision of 96.0%, average recall of 93.0%, and mean average precision (mAP) of 95.0%, as shown in Table 1, which shows that our proposed model achieved the highest accuracy score than other SOTA models. Visual results of the proposed modified YOLOv5s are shown in Figure 9. Furthermore, the inherent capabilities of UAVs, coupled with the YOLOv5s model evaluated in this study, offer a reliable solution for real-time pest detection, showcasing significant potential to optimize and enhance agricultural production within a drone-centric ecosystem.

## 5. Conclusions

In our study evaluating nine object recognition models, the standout performer was our specialized UAV-oriented model. Throughout the experiments, conducted with a meticulously curated dataset featuring five distinct insect species, ants, grasshoppers, palm weevils, shield bugs, and wasps, our model consistently outshone its counterparts in accuracy, recall, and mean average precision (mAP). What sets our model apart is not just its impressive performance but also its efficiency, demonstrating superior inference capabilities while demanding fewer computational GFLOPs and maintaining a more manageable model size. This positions our proposed model as a robust solution for real-time species detection, highlighting its prowess in the context of UAV and technology integration.

Moreover, our model excels in enhanced crop monitoring using UAVs, demonstrates proficiency in handling high-resolution data for yield prediction, and proves effective in detecting and preventing crop diseases.

## Figures and Tables

**Figure 1 plants-13-00653-f001:**
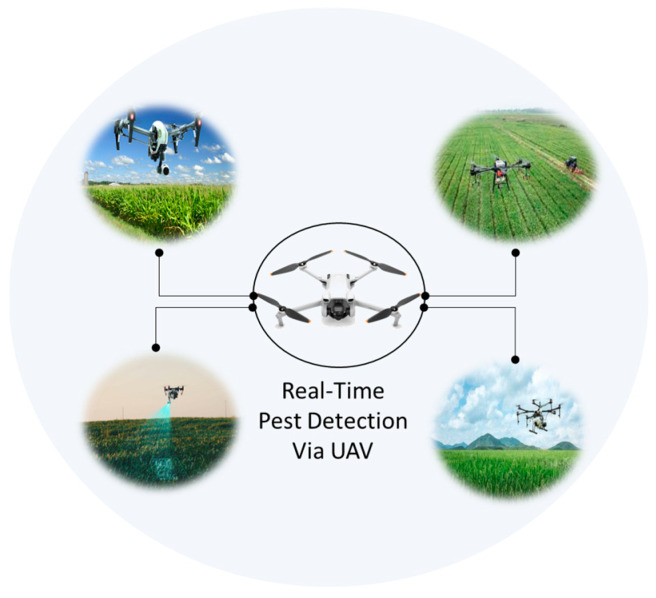
UAV-based early pest detection system to assist agricultural department.

**Figure 2 plants-13-00653-f002:**
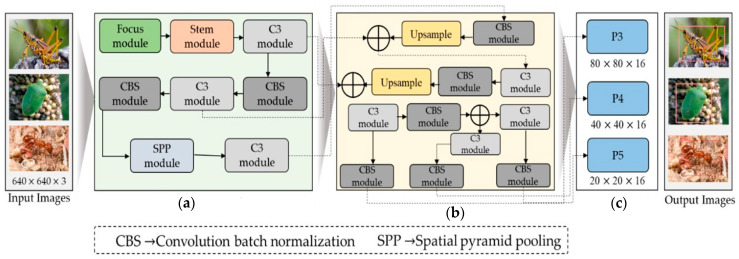
The generic overview of the proposed work for pest detection comprises the following components: (**a**) the backbone module, (**b**) the neck module, and (**c**) the output head module. “Focus module” and “Stem module” presented in green and pink box are our modification blocks in YOLO.

**Figure 3 plants-13-00653-f003:**
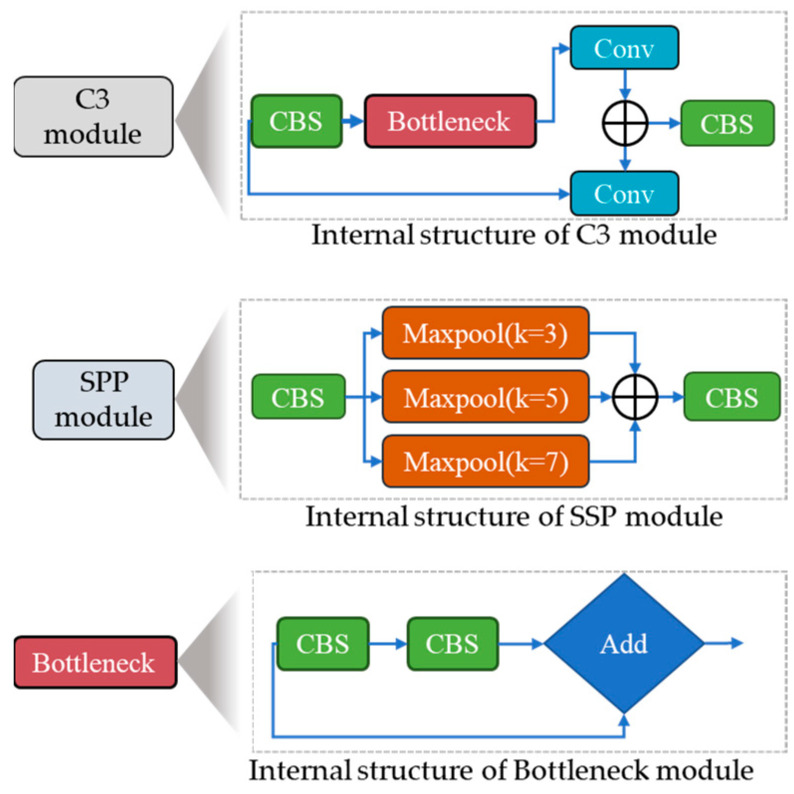
Various modules architecture used in the proposed YOLO for pest detection.

**Figure 4 plants-13-00653-f004:**
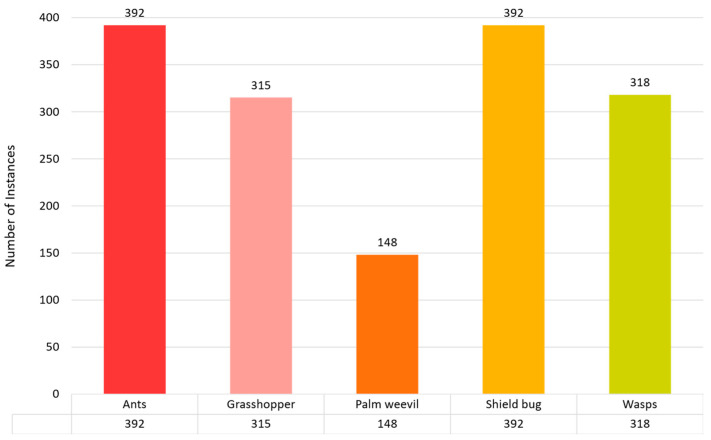
Pest or insect detection dataset distribution for training purposes.

**Figure 5 plants-13-00653-f005:**
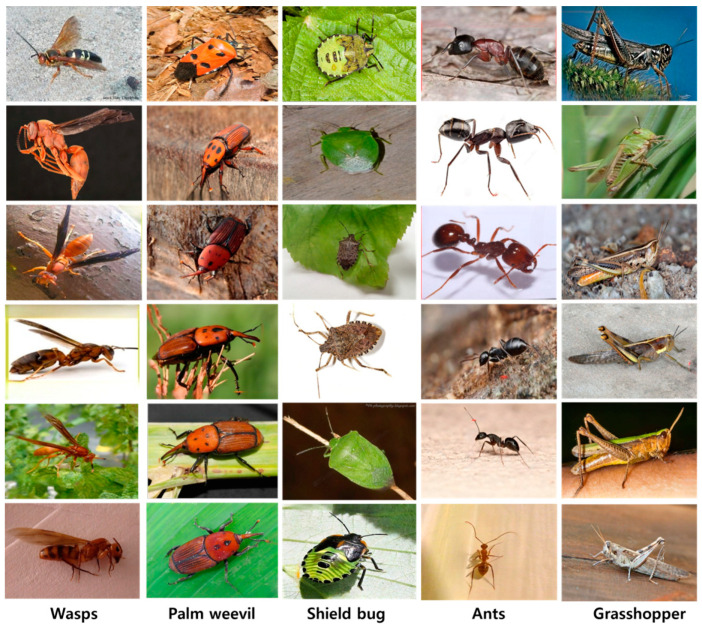
Various samples of pests for each dataset class.

**Figure 6 plants-13-00653-f006:**
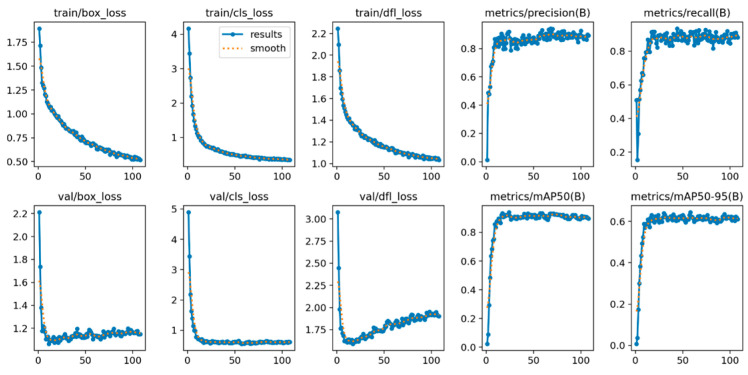
The X-axis shows the number of epochs, and the Y-axis shows the corresponding score of each evaluation matrix. This shows the model’s efficacy using various evaluation metrics.

**Figure 7 plants-13-00653-f007:**
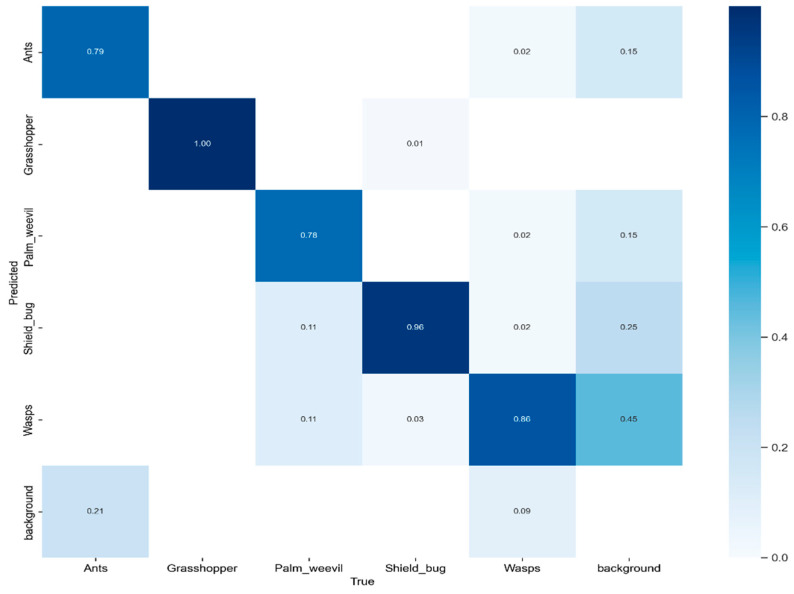
The proposed model’s confusion matrix using a self-created dataset.

**Figure 8 plants-13-00653-f008:**
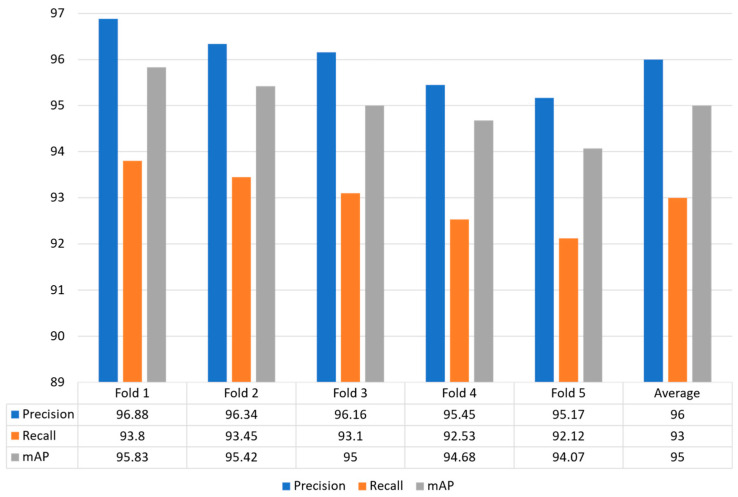
Illustrated the model generalization capability using 5-fold cross validation.

**Figure 9 plants-13-00653-f009:**
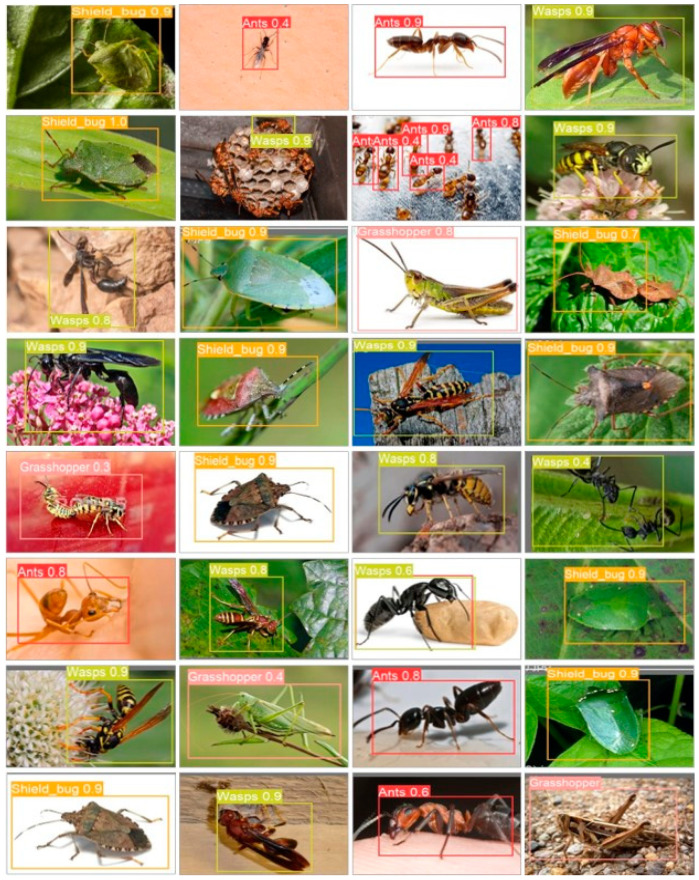
The visual results of the proposed model, which show the model effective analysis.

**Table 1 plants-13-00653-t001:** Analysis of the proposed model with Faster-RCNN and various versions of the YOLO models.

Models	Classes	Precision	Recall	mAP
Faster-RCNN	Ants	0.73	0.74	0.76
	Grasshopper	0.98	0.99	1.00
	Palm_weevil	0.99	0.88	0.98
	Shield_bug	0.96	0.97	1.00
	Wasp	0.94	0.86	0.91
	Average	0.92	0.89	0.92
YoloV3	Ants	0.59	0.75	0.64
	Grasshopper	0.86	0.84	0.91
	Palm_weevil	0.91	0.93	0.95
	Shield_bug	0.88	0.91	0.93
	Wasp	0.87	0.9	0.88
	Average	0.82	0.87	0.86
YoloV4	Ants	0.65	0.76	0.71
	Grasshopper	0.87	0.83	0.93
	Palm_weevil	0.93	0.92	0.96
	Shield_bug	0.9	0.94	0.95
	Wasp	0.88	0.91	0.89
	Average	0.85	0.87	0.89
YOLOv5n	Ants	0.57	0.74	0.68
	Grasshopper	0.92	0.87	0.94
	Palm_weevil	1.00	0.98	0.99
	Shield_bug	0.93	0.97	0.98
	Wasp	0.92	0.82	0.87
	Average	0.87	0.88	0.89
YOLOv5s	Ants	0.80	0.68	0.78
	Grasshopper	0.90	0.87	0.88
	Palm_weevil	0.96	1.00	0.99
	Shield_bug	0.97	0.91	0.98
	Wasp	0.89	0.71	0.87
	Average	0.91	0.83	0.90
YOLOv5m	Ants	0.86	0.65	0.73
	Grasshopper	0.95	1.00	0.99
	Palm_weevil	1.00	0.82	0.99
	Shield_bug	0.94	0.93	0.97
	Wasp	0.93	0.82	0.84
	Average	0.94	0.84	0.91
YOLOv5l	Ants	0.72	0.72	0.98
	Grasshopper	1.00	0.93	0.99
	Palm_weevil	0.76	1.00	0.98
	Shield_bug	0.89	0.96	0.96
	Wasp	0.88	0.84	0.89
	Average	0.85	0.89	0.92
YOLOv5x	Ants	0.70	0.72	0.76
	Grasshopper	0.97	1.00	0.99
	Palm_weevil	1.00	0.87	0.97
	Shield_bug	0.97	0.98	0.99
	Wasp	0.92	0.84	0.89
	Average	0.91	0.88	0.92
EPD	Ants	0.79	0.76	0.80
	Grasshopper	0.98	1.00	0.99
	Palm_weevil	1.00	0.89	0.98
	Shield_bug	0.97	0.98	0.99
	Wasp	0.94	0.86	0.90
	Average	0.94	0.90	0.93
Our model	Ants	0.86	0.83	0.84
	Grasshopper	0.99	1.00	0.99
	Palm_weevil	1.00	0.94	0.99
	Shield_bug	0.99	0.98	0.99
	Wasp	0.96	0.89	0.94
	Average	0.96	0.93	0.95

**Table 2 plants-13-00653-t002:** Comparative Analysis of our model with different YOLOv5 versions in terms of model size, GFLOPs, and FPS using CPU.

Model	Model Size	GFLOPs	FPS (CPU)
YOLOv5n	3.65	4.20	31.02
YOLOv5s	14.1	16.00	21.25
YOLOv5m	40.2	48.30	10.16
YOLOv5l	88.5	108.30	6.62
YOLOv5x	165	204.70	3.90
Our model	14.0	15.00	22.75

## Data Availability

Data is contained within the article.

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
