# Peer review of "AI-Enabled Crop Management Framework for Pest Detection Using Visual Sensor Data"

_plants, 2024, doi:10.3390/plants13050653_

Round 1

Reviewer 1 Report

Comments and Suggestions for Authors

Summary

presents a novel approach to pest detection in agriculture by integrating UAV technology and AI. The authors propose a modified YOLOv5s model to effectively identify and categorize pests in visual sensor data captured by drones. The study evaluates nine object recognition models and demonstrates the superior performance of the specialized UAV-centric model in terms of accuracy, recall, and mean average precision (mAP). The proposed model not only exhibits impressive performance but also demonstrates efficiency, requiring fewer computational resources and maintaining a manageable model size. The integration of high-resolution data for yield prediction further enhances the reliability of the model. The paper emphasizes the potential of the proposed framework to revolutionize pest management, maximize agricultural yields, and contribute to sustainable agriculture. Additionally, the authors provide insights into the methodology, experimental results, and the implications of their research for the agricultural sector.

Major:

·       Model evaluation.

Since the number of samples is not very large, it would be better to conduct cross-validation or select multiple validation/test sets to evaluate the models.

Minor:

·       Highlight the evaluation score.

In Table 2, the best performance of the model should be highlighted to enhance table readability.

·       Code availability.

It would be beneficial if the code were made publicly available.

·       Distribution of the dataset among the five categories.

Since Figure 4 and Table 1 provide the same information, it would be beneficial to merge them or retain just one of them.

Author Response

Thank you for reviewing our manuscript. Your constructive comments have provided valuable insights that will help us enhance the quality and impact of our work

Reviewer 2 Report

Comments and Suggestions for Authors

Line 2

Delete drone assisted (If I understood correctly, the presented research does not test the use of a complete UAV system equipped with YOLOv5s in the field, but is the study only of a part: a YOLOv5s-modified insect recognition system)

Lines 28, 29,30

Delete. The inherent capabilities of UAVs, combined with our enhanced model, offer a reliable solution for real-time pest detection, demonstrating significant potential to optimize and improve agricultural production within a drone-centric ecosystem.

Replace with. The inherent capabilities of UAVs, combined with the YOLOv5s model tested here, could offer a reliable solution for real-time pest detection, demonstrating significant potential to optimize and improve agricultural production within a drone-centric ecosystem.

Lines 34 and 35

Delete agricultural sector is vital for economic enhancement, emphasizing the need to identify harmful pests.

Replace with agricultural sector is vital to economic enhancement, and it is essential to identify the pests that harm it.

Line 38

Delete imperative.

Replace with important.

Lines 118-119-120

Delete: The agricultural sector ensures a nation's overall security and socio-economic development. Hence, it is necessary to identify harmful pests in the natural environment. Therefore…(it’s a repetition)

Line 451

Delete                   UAV-centric model.

Replace with         UAV oriented model.

Author Response

Thank you for reviewing our manuscript. Your constructive comments have provided valuable insights that will help us enhance the quality and impact of our work.

Round 2

Reviewer 1 Report

Comments and Suggestions for Authors

The authors have addressed all comments.

Author Response

Thank you for your thoughtful review of the paper; your feedback is greatly appreciated.